# The Overlap of Allergic Disorders and Upper Gastrointestinal Symptoms: Beyond Eosinophilic Esophagitis

**DOI:** 10.3390/nu17081355

**Published:** 2025-04-16

**Authors:** Oksana Wojas, Edyta Krzych-Fałta, Paulina Żybul, Marta Żalikowska-Gardocka, Tomasz Ilczuk, Konrad Furmańczyk, Bolesław Samoliński, Adam Przybyłkowski

**Affiliations:** 1Department of Prevention of Environmental Hazards, Allergology and Immunology, Medical University of Warsaw, 02-097 Warsaw, Poland; konrad.furmanczyk@wum.edu.pl (K.F.); boleslaw.samolinski@wum.edu.pl (B.S.); 2Department of Basic Nursing, Medical University of Warsaw, 01-445 Warsaw, Poland; 3Department of Gastroenterology and Internal Medicine, Medical University of Warsaw, 02-097 Warsaw, Poland; paulina.zybul@wum.edu.pl (P.Ż.); marta.zalikowska-gardocka@wum.edu.pl (M.Ż.-G.); adam.przybylkowski@wum.edu.pl (A.P.); 4Department of Pathology, Medical University of Warsaw, Pawińskiego 3B, 02-106 Warszawa, Poland; tomasz.ilczuk@wum.edu.pl; 5Institute of Information Technology, Warsaw University of Life Sciences, Nowoursynowska 166, 02-787 Warsaw, Poland

**Keywords:** eosinophilic esophagitis, dysphagia, asthma, allergic rhinitis, atopic dermatitis, food allergies, epithelial barriers, eotaxin 1, desmoglein 1

## Abstract

Eosinophilic esophagitis (EoE) is a chronic disease which clinically presents with symptoms related to esophageal dysfunction, while pathologically it is characterized by eosinophilic infiltration of esophageal epithelium. Most patients with EoE present with food and/or inhalant allergy symptoms. The results of animal model studies and genetic studies, as well as the efficacy of elimination diets in managing the symptoms, suggest an atopic background of the disease. The aim of this study was to evaluate the prevalence of EoE in a group of patients with upper gastrointestinal symptoms and food and/or inhalant allergies and to assess the influence of drugs used in type I allergies on the results of endoscopic, histopathological, and immunohistochemical tests. **Methods**: This was a prospective observational study. Patients with inhalant/food allergies and upper esophageal symptoms constituted the study group while patients without allergies who were diagnosed with dyspepsia or irritable bowel syndrome constituted the control group. All study group subjects underwent allergy testing, including prick testing and blood tests. All participants underwent a gastroscopy with specimen collection. Esophageal specimens were stained for eotaxin-1 and desmoglein-1. **Results**: Based on histopathology results, eosinophilic esophagitis was found in 9 of the 73 patients from the study group. All patients with EoE presented with multimorbidity and were diagnosed with at least one allergic disease in addition to EoE. Positive staining for CCL-11 was found in 56 (78%) patients in the study group, including all patients with EoE while only 3 (17%) individuals from the control group showed positive staining. The presence of DSG-1 in esophageal specimens was detected in 6 (7%) subjects from the study group in contrast to 14 (78%) subjects from the control group. DSG-1 was not found in any of the specimens of patients diagnosed with EoE. **Conclusions**: EoE is a rare disease, usually accompanied by allergic multimorbidity. Positive staining for eotaxin-1 and negative staining for desmoglein-1 in patients with esophageal symptoms and allergies but who did not meet EoE diagnostic criteria could be indicative of subclinical course of the disease or a masking effect of corticosteroids. It is now vitally important for both researchers and practicing clinicians to recognize that eosinophilic esophagitis (EoE) is not a homogeneous disease but rather consists of multiple subtypes (phenotypes). The so-called “classic” form of EoE—defined by current diagnostic criteria as the presence of more than 15 eosinophils per high power field on histopathological examination—appears to represent only the tip of the iceberg. There is an urgent need for further research in order to refine endoscopic techniques, expand the scope of histopathological assessments, and identify novel biomarkers to better define the distinct phenotypes of eosinophilic esophagitis.

## 1. Introduction

Eosinophilic esophagitis is a chronic disease which clinically presents with symptoms related to esophageal dysfunction while pathologically it is characterized by eosinophilic infiltration of esophageal epithelium [1,2]. In the 1980s, eosinophilic infiltration of the esophagus was thought to be associated with gastroesophageal reflux disease (GERD). In 1989, Attwood et al. in their study entitled “Oesophageal Asthma—an episodic dysphagia with eosinophilic infiltration” identified a subgroup of 15 patients with a different disease course among the total group of 100 adults with GERD. Those patients presented with dysphagia and were found to have normal pH-metry results along with a significantly more marked eosinophilic infiltration of the esophageal epithelium compared to the other subjects [3]. Further research into this phenomenon resulted in the publication of a study titled “Esophageal Eosinophilia with Dysphagia. A distinct clinicopathological syndrome” in 1993, in which a new entity, known to this day as eosinophilic esophagitis, was defined by the same authors. The study described the clinicopathological profile of the condition for the first time, which consists of dysphagia, normal pH-metry, and a significant eosinophilic infiltrate in the esophagus (more than 15 cells per high power field). Since that date, nearly 30 years later, the number of publications and studies on eosinophilic esophagitis has increased significantly [4].

Today, eosinophilic esophagitis ranks high in prevalence among upper gastrointestinal diseases and constitutes a significant public health problem worldwide. This is probably related, on the one hand, to the growing awareness of EoE among clinicians and, on the other hand, to the rapid development and greater availability of diagnostic tests, particularly endoscopic techniques [5]. It should be noted that intensive research has recently contributed to the understanding that eosinophilic esophagitis is not a homogeneous disease. Esophageal eosinophilia, which has been considered the basis of diagnosis of this disease for years, has been questioned and may be less important than previously thought. There are phenotypes of this disease that are independent of eosinophils (EoE beyond eosinophils). There is a high probability that EoE is in some way the tip of the iceberg or is the most extreme phenotype with several subtypes (variants) of this disease [6,7]. The objective of this study was to evaluate the esophageal mucosa in a group of patients with inhalant and food allergies and esophageal symptoms.

## 2. Materials and Methods

### 2.1. Ethical Statements

The protocol of the study was approved by the Bioethics Committee of the Medical University of Warsaw (decision no. KB/155/2017).

### 2.2. Study Participants

This prospective study was conducted at the Department of the Prevention of Environmental Hazards and Allergology and the Department of Internal Medicine and Gastroenterology of the Medical University of Warsaw from 2017 to 2022. The inclusion criteria for the study group were as follows: age between 18 and 70, allergy diagnosed by skin prick tests (SPTs) and/or sIgE test, presence of gastrointestinal symptoms (aphagia, dysphagia, odynophagia, heartburn, chest pain, epigastric pain, nausea, vomiting). A positive result for SPTs was defined as presence of a wheal of larger than 3 mm and erythema of more than 5 mm in diameter, while for sIgE testing, a grade 3 response (3.5–17.5 kU/L) was considered positive. We adopted the following exclusion criteria: esophageal eosinophilia as a result of conditions other than EoE such as esophageal achalasia, gastroesophageal reflux disease, Crohn’s disease, hypereosinophilic syndrome, connective tissue diseases, celiac disease, pemphigoid, infections (Candida, Herpes), drug allergies, and history of transplantation (graft versus host disease).

The sole inclusion criterion for the control group was the diagnosis of dyspepsia. The following exclusion criteria were adopted: esophageal achalasia, gastroesophageal reflux disease, Crohn’s disease, hypereosinophilic syndrome, connective tissue diseases, celiac disease, pemphigoid, infections (Candida, Herpes), drug allergies, and history of transplantation (graft versus host disease), diagnosis of allergy based on clinical history and negative results of specific IgE tests for inhalant and food allergens. All study participants signed an informed consent to participate in the study.

### 2.3. Allergy Evaluation

In all study participants, skin prick tests (SPTs) were performed with Allergopharma-Nexter (Hermann-Körner-Straße 52-54, 21465 Reinbek, Germany) kits for inhalant allergens (birch, alder, hazel, grass mix, rye, dog, cat, Dermatophagoides farinae, Dermatophagoides pteronyssinus, Alternaria, Aspergillus, common mugwort, ribwort plantain) and food allergens (cow’s milk, wheat, hen’s egg, soy, hazelnut, fish (cod)). Testing for serum sIgE was also carried out for the above-mentioned inhalant and food allergens (Polycheck, BiocheckGmbH, Vorbergweg 41, 48159 Münster, Germany). All patients completed a questionnaire outlining their allergy symptoms, gastrointestinal symptoms, past medical history, and medication history. Patients’ current medications were assessed with special focus on systemic, inhaled, or intranasal corticosteroids (CSs), antihistamines (AHs), and proton pump inhibitors (PPIs).

### 2.4. Endoscopy, Histopathology, and Immunohistochemical Staining

In all study participants, esophagogastroduodenoscopy (camera type GIF-H190, Olympus Deutschland GmbH, Amsinckstraße 6320097 Hamburg, Germany) was performed, which involved the collection of six specimens from the upper, middle, and lower esophagus.

We carried out hematoxylin and eosin staining, along with immunohistochemical testing for the presence of eotaxin-1 (CCL11) and desmoglein-1 (DSG-1). The following diagnostic criteria for EoE were applied [1,2]:Min 15 eosinophils per high power field at 400× magnification in histopathological testing (specimens obtained from endoscopic biopsy);Exclusion of other causes of esophageal eosinophilia.

Immunohistochemical reactions were performed on 4 µm paraffin samples, using peroxidase activity in an enzymatic reaction. After deparaffinization and rehydration, the preparations underwent an epitope retrieval process. To detect the examined proteins, primary antibodies were used, such as Desmoglein 1 Monoclonal Antibody (27B2), catalog no. 32-6000 (Thermo Fisher Scientific, 168 Third Avenue Waltham, MA 02451, USA); and Eotaxin Monoclonal Antibody (43911) catalog no. MA5-23831 (Thermo Fisher Scientific, 168 Third Avenue Waltham, MA 02451, USA). The samples were then transferred to a 3% hydrogen peroxide solution to block endogenous peroxidase activity. A 5% serum donkey solution was used to block non-specific antibody binding sites. Next, the samples were incubated in a primary antibody solution. Secondary antibodies conjugated directly with peroxidase particles (catalog no. DS9800, Leica Biosystems, Heidelberger Str. 17-19, 69226 Nußloch, Germany) were used to detect the primary antibodies. The reaction was visualized using 3,3′-diaminobenzidine as chromogen. Cell nuclei were visualized after contrast staining with hematoxylin.

For an objective assessment of the immunohistochemical reaction intensities, we used the immunoreactive score (IRS) scale according to Remmele and Stagner. This is a semi-quantitative scale incorporating the percentage of positive cells and staining intensity in five visual fields of the light microscope at 200× magnification. The final IRS is a product of the percent positive cells (0, no cells with positive reaction; 1, ≤10% cells with positive reaction; 2, 11–50% cells with positive reaction; 3, 51–80% cells with positive reaction; 4, >80% cells with positive reaction) and staining intensity (0, no color reaction; 1, poor color reaction; 2, moderate color reaction; 3, intensive color reaction), ranging from 0 to 12 (0–2, poor reaction; 3–5, moderate reaction; 6–12, intensive reaction).

### 2.5. Statistical Analyses

In statistical analyses, frequencies for individual allergens and diseases were calculated, and significant differences in proportions (frequencies) were compared between two main groups of patients (healthy and sick) and three groups of allergic patients. A 2-sample test for equality of proportions with continuity correction and a 3-sample test for equality of proportions without continuity correction were used for comparison. The significance level of α = 0.05 was adopted, meaning that differences were considered significant for *p*-values < 0.05. Since 95 tests were performed for explanatory hypotheses (associations without pre-established hypotheses), the Bonferroni correction was applied to avoid type I error inflation. After the Bonferroni correction for 95 hypotheses, the adjusted significance level was 0.0005.

All calculations were made using the R statistical package, version 3.2 (www.r-project.org).

## 3. Results

A total of 73 patients with allergies (28 males [38%] and 45 females [62%]) aged 18 through 70 years (mean age 36.3 years), all of whom presented with upper gastrointestinal symptoms, constituted the study group (Figure 1). The most common symptoms reported by patients, which were present in all subjects in this group, were heartburn, epigastric pain, and dysphagia. The control group consisted of 13 males (72%) and 5 females (28%) aged between 18 and 70 years (mean age 52.1) (Table 1). Histopathological criteria for eosinophilic esophagitis (EoE) were met in 9 (12%) patients; 1 patient (1%) was diagnosed with lymphocytic esophagitis. Positive staining for CCL-11 was observed in 56 (78%) patients in the study group, including all patients with EoE and 3 (17%) from the control group (*p* = 0.000461). DSG-1 was visualized in esophageal sections from 6 (7%) patients in the study group and was not observed in any of the patients with EoE; it was found in 14 (78%) patients from the control group (*p* = 0.0000) (Table 1). Based on skin prick tests (SPTs) results, the most common allergies were to trees including alder, birch, and hazel, as well as allergies to grasses. House dust mite allergies accounted for a slightly lower percentage. Regarding food allergens, the most common were hazelnuts and wheat. Very similar results were obtained from the sIgE testing, with allergies to trees, grasses, and house dust mites being the most common among the inhalant allergens panel, and wheat being the most common allergen from among the food panel (Table 2).

It was observed that intranasal and inhaled corticosteroids can help alleviate EoE symptoms if swallowed. In order to explore the potential effects of allergy treatment on esophageal mucosa, we divided the study group patients into three subgroups based on histopathology results.

Patients diagnosed with EoE based on histopathological examination (15 or more eosinophils per high power field), with positive staining for CCL-11 and negative staining for DSG-1: 9 patients. In this subgroup, one patient was not taking any corticosteroids, six patients were on both ICS and InCS, and two patients were only on InCSs. In this group, 89% of patients were diagnosed with allergies to house dust mites and birch, 33% to grasses, hazel and alder, 100% to hazelnuts, 78% to wheat, and 56% to cow’s milk.

Patients with unconfirmed EoE on histopathological examination (less than 15 eosinophils per high power field), with positive staining for CCL-11 and negative staining for DSG-1: 47 patients. In this subgroup, 14 patients were taking ICSs, 44 patients were taking InCSs, and 8 patients were on both ICSs and InCSs. One patient in the group was receiving systemic corticosteroids. In this group, 85% of patients were found to be allergic to grasses, 55% to house dust mites, 66% to birch, 68% to alder, and 64% to hazel. Moreover, 10% of patients were allergic to wheat and milk and 9% to hen’s egg.

Patients with unconfirmed EoE on histopathological examination and negative staining for CCL-11: 17 patients. Within this subgroup, 11 patients were found to have negative staining for DSG-1 while positive staining results were obtained in 6 patients. In this subgroup, 14 patients were not given any corticosteroid therapy, while 2 patients were taking InCSs, and 1 patient was taking ICSs. In this group, 56% were allergic to hazel, 50% to birch, 38% to alder and cat, 25% were allergic to house dust mites, and 56% were allergic to hazelnuts.

We also performed a logistic multiple regression model in two variants, where the response variable was EoE diagnosis and positive eotaxin-1 staining, adjusted for age, sex, type of allergy, DSG, ICS, and INCS. However, sex, age and type of allergy were not significant factors, which confirms that the results of previous statistical analyses were not biased by the influence of these confounders.

## 4. Discussion

The purpose of this study was to evaluate the properties of esophageal mucosa in patients with inhalant and food allergies and esophageal symptoms. Eosinophilic esophagitis is usually initially diagnosed by gastroenterologists, with allergological diagnostics implemented only in the second stage. In the present study, a decision was made to take a different approach in order to verify the incidence of this disease in patients originally diagnosed with allergies and to examine its relationship to the multimorbidity of allergic diseases. Eosinophilic esophagitis (EoE) is a disease of global prevalence. Most data on the disease come from the United States, Europe, and Australia, with slightly fewer data originating from Asia. Africa and India are notable exceptions as no cases of EoE have been documented in these regions to date [8,9]. Based on the epidemiological data, a profile of a “typical” patient with EoE was established as a male Caucasian under 50 years of age, residing in a rural area in a cool and dry climate, and having a positive history of atopy [1,8,9,10].

In this study, eosinophilic esophagitis was diagnosed in 12% of patients in a group of 73 individuals with inhalant and food allergies. While a higher prevalence in this group might have been expected, this represents a relatively significant finding within our specific cohort. We also found that EoE was more common in male patients as it was diagnosed in seven men compared to two women. All patients were in the fourth decade of their lives.

The incidence of the disease has increased significantly over the past 30 years. In the 1990s, it was considered a rare disease, whereas today it ranks second in incidence among upper gastrointestinal diseases, after gastroesophageal reflux disease. The prevalence of EoE varies between countries and continents [8]. In a Danish study, Dalby et al. estimated the incidence of EoE in the pediatric population at 2.3/100,000; in comparison, in their study carried out in Ohio, Noel et al. reported on an incidence of 90.7/100,000 [11,12]. Based on a meta-analysis, Arias et al. estimated the prevalence of the disease at 22.7/100,000, the rate being higher in adults (43.4/100,000) than in children (29.5/100,000) [13]. In addition to geographic differences in the incidence of EoE, the clinical profiles of the study groups appear to be of importance. Thus, when endoscopic screening of the upper gastrointestinal tract is performed, EoE is detected in 2.4–6.6% of patients, whereas in cases of endoscopic examinations being performed in patients with dysphagia, the percentage is higher, ranging from 12% to 23% [14,15].

In the study group, the most common symptom reported by patients was heartburn (70%), followed by epigastric pain (55%) and dysphagia (47%). In contrast, in the EoE group, dysphagia was the most prominent and common symptom, with chest pain, nausea, and heartburn being less common. According to an analysis of the available literature, the leading clinical symptom in adult patients with EoE is dysphagia, which occurs in 60–100% of patients [1,9]. On the other hand, episodes of foreign body (food bolus) entrapment within the esophagus occur in 25% of patients [8,16]. In this study, food boluses entrapped within the esophagus were found in two patients, both in the EoE group. According to the literature, chest pain occurs in 30–60% of patients [1,15,16]. In contrast, only 4% of patients with abdominal pain and 6% of patients with chest pain are diagnosed with EoE [1,5]. Symptoms such as abdominal pain, heartburn, belching, nausea, vomiting, and diarrhea are not pathognomonic of EoE. It is also believed that no presentation pathognomonic of EoE can be identified from endoscopic examination of the esophagus. In some patients, the esophagus may be unremarkable on endoscopic examination while effusions, trachealization, rings, strictures, cat’s esophagus, and erosions may also be present [16]. In most of our patients, the esophagus was unremarkable on endoscopic examination. Most patients with EoE presented with morphological changes on endoscopic examination (trachealization in six patients, narrowing in two patients, white plaques in one patient, and normal esophagus in two patients).

In the study group, the most common allergic disease was allergic rhinitis (AR) observed in 94% of patients. Food allergy was found in 73% and asthma in 27% of patients. In addition, 72% of patients presented with AR concomitant with food allergy and 26% had concomitant asthma. Only one patient with isolated food allergy was diagnosed with EoE while for the remaining eight patients, the diagnosis of EoE was related to allergic comorbidities. All patients were diagnosed with AR with concomitant food allergy, asthma, and/or atopic dermatitis. In recent years, the topic of multimorbidity—that is, one or more diseases occurring in a comorbid fashion with the primary disease—and, consequently, the topic of multidirectional, multispecialty patient care, have been widely discussed in allergology and other areas of medicine. In 2015, Bousquet et al. described the phenomenon of multimorbidity in allergic diseases, allergic rhinitis, asthma, and atopic dermatitis, noting the similar immunological and non-immunological pathomechanisms of these conditions [17,18,19]. The association of allergic diseases with the occurrence of eosinophilic esophagitis was confirmed in numerous studies. Atopy is present in 20–80% of patients with EoE [20]. In addition, 50% of patients with EoE have atopy and a family history of allergy [14,20,21,22]. An association between EoE and other allergic diseases was shown in several studies: 25–50% of patients with EoE have asthma, 30–90% have AR, and 10–25% have AD. As much as 10–25% of patients with EoE have food allergies compared to 8% in the general population [9,21]. Hill et al. evaluated the prevalence of eosinophilic esophagitis in a group of patients with IgE-mediated food allergies and found that EoE in this group occurred in 4.7% of patients, compared to 0.45% in the general population [22]. A year later, in a subsequent paper, the same authors demonstrated a strong association between EoE and allergic rhinitis and found that exposure to inhaled allergens exacerbated EoE [23]. In a study group of EoE patients, Ram et al. observed that the disease was exacerbated in 14% of cases during exposure to seasonal allergens, which was confirmed by esophageal biopsy results [24]. As demonstrated by Fahey et al., the frequency of diagnosis of new EoE cases increases in the spring and summer seasons, which is associated with exposure to seasonal allergens [25]. In contrast, Larsson et al. showed that the number of patients with an episode of a foreign body (food bolus) entrapped in the esophagus is higher during the pollination seasons in summer and fall [26]. In a single-center retrospective study, Zdanowicz et al. found that 8.31% of a group of 433 patients had presented with EoE. In the group of EoE patients, food allergy was confirmed in 11.11% of cases while inhalant allergy was confirmed in 23.08% of cases [27]. Straumann and Geuter emphasize that while allergy probably plays an important role in the pathogenesis of EoE, the disease is multifactorial and other factors, such as epithelial barrier disorders, play an important role in its development [28]. In our study, in a group of 73 subjects, allergy to grasses was found in 63%, allergy to trees was found in 60%, and allergy to house dust mites was found in 54% of patients, based on SPT and sIgE testing. On the other hand, eight individuals within the EoE group were found to be allergic to house dust mites, while three were also allergic to grasses, trees, and cats. Only one patient was found to have no inhalant allergies. In the study group, the most common food allergen was hazelnut (30%), followed by wheat (18%) and milk (14%). In the EoE group, all patients were found to be allergic to hazelnuts, while five patients were found to be allergic to cow’s milk proteins. One patient was found to be allergic to soy, one to fish, and seven to wheat.

Immunohistochemical assays were carried out in the study group of patients with allergy and gastrointestinal symptoms to reveal positive staining for eotaxin-1 (CCL-11) in 56 (78%) patients, including all patients with eosinophilic esophagitis. In contrast, positive staining was obtained in only three (17%) subjects in the control group. Eotaxin-1 is a chemokine with strong chemotactic and eosinophil-activating effects. It is the key chemokine responsible for eosinophilic colonization of lung bronchioles in asthma patients. It is highly likely that a similar role is played by this chemokine in eosinophilic esophagitis [29,30,31,32]. Positive staining for eotaxin-1 within the mucosa of patients with allergy and gastrointestinal symptoms, in whom EoE had not been confirmed, may suggest subclinical disease or some propensity to develop eosinophilic inflammation within the mucosa.

The majority of patients within the study group—67 (91%), as well as all patients diagnosed with EoE, were found to lack desmoglein-1 (DSG-1) expression, whereas relevant staining was positive in most patients in the control group. The group of patients with allergy and symptoms related to the upper gastrointestinal tract, as well as positive immunohistochemical results for eotaxin and negative results for desmoglein, is the most interesting. Although histopathological examination did not confirm EoE in patients from this group, we believe that the positive staining for eotaxin may indicate eosinophilic inflammation, while the negative staining for desmoglein could suggest damage to the integrity of the epithelial barrier. Therefore, we are likely dealing with one of the phenotypes of EoE in this case. Desmoglein-1 is a cadherin involved in the formation of desmosomes, which are responsible for intercellular connections within the epithelium [33,34,35,36,37,38,39]. Patients with eosinophilic esophagitis are likely to present with reduced levels of desmoglein-1 in the epithelium, resulting in the disruption of epithelial integrity and weakening of the epithelial barrier within the esophagus [34,40,41,42,43,44,45,46]. Other environmental factors predisposing individuals to EoE include preterm birth, cesarean delivery, formula feeding, intrauterine infections, early antibiotic exposure, and a high animal fat diet (the so-called Western diet), which leads to dysbiosis [37,38]. Celebi Sozener et al. highlighted the epithelial barrier hypothesis, which provides a mechanistic explanation of how these external (environmental) factors contribute to the rapid rise in allergic and autoimmune diseases [37].

Global environmental factors such as climate change, microplastic exposure, tobacco smoke, biodiversity loss, and dietary shifts due to urbanization, modernization, and globalization are responsible for epithelial barrier impairment. Consequently, they contribute to the increasing incidence of allergic diseases, including eosinophilic esophagitis [38,42]. In their exceptionally insightful study, Barchi et al. emphasized that endoscopic examination is fundamental in diagnosing eosinophilic gastrointestinal diseases (EGIDs). Furthermore, the implementation of virtual chromoendoscopy techniques and artificial intelligence (AI) in result analysis may prove highly beneficial. Histopathology remains a cornerstone in both the diagnosis and management of EGIDs. The assessment of eosinophilic infiltration in tissue sections continues to be the gold standard for diagnosing these conditions [47].

In the context of eosinophilic gastritis (EoG) and eosinophilic enteritis (EoN), there is an urgent need for clearly defined diagnostic criteria and a better understanding of the correlation between eosinophilic infiltration and clinical manifestations. Future research directions should focus on a comprehensive analysis not only of eosinophils but also of other microscopic changes and inflammatory cells involved in the disease process. Establishing robust histopathological criteria for eosinophilic esophagitis (EoE) will require detailed investigations beyond eosinophilic infiltration alone [47].

Advancements in artificial intelligence and mechanistic approaches hold great promise for revolutionizing histopathological analysis, as AI-driven systems are increasingly capable of detecting even subtle pathological patterns. Furthermore, a deeper understanding of the molecular aspects of EGIDs will facilitate personalized medical approaches. Traditional histopathological methods should be complemented by molecular profiling and biomarker studies to develop more targeted therapies tailored to individual patient profiles.

The synergy between endoscopy and histopathology is fundamental for the diagnosis and management of EGIDs. Continuous integration and refinement of these two diagnostic modalities offer hope for expanding our understanding of eosinophilic diseases, ultimately leading to improved diagnostic accuracy and treatment outcomes [47]. Another key aspect of EoE pathogenesis is epithelial barrier dysfunction, which facilitates the penetration of allergens and perpetuates the inflammatory response. Numerous environmental factors, including diet, cesarean delivery, childhood antibiotic use, formula feeding, early exposure to proton pump inhibitors (PPIs), cold climate, and poor living conditions, have been associated with an increased risk of EoE [48]. Conversely, some studies suggest that childhood exposure to furry pets and *Helicobacter pylori* infection may be linked to a reduced risk of developing EoE [48,49,50].

The role of the gut microbiota in immune regulation is increasingly recognized as significant [51]. Therapeutic strategies aimed at modulating microbiota composition to alleviate inflammation and allergen sensitization may hold promise for EoE treatment [51,52,53,54,55].

A study by Zhang et al. found an increased abundance of *Haemophilus* and *Aggregatibacter* species, along with a decrease in *Firmicutes*, in patients with active EoE [56]. Meanwhile, research by Arias et al. demonstrated that early interactions between epithelial cells and the esophageal microbiota could modulate *CXCL16* expression and recruit invariant natural killer T (iNKT) cells to the esophageal epithelium [57]. In a mouse model, the administration of *Lactococcus lactis* NCC 2287 induced histopathologic remission of EoE [58].

The absence of desmoglein expression in patients with EoE and some patients in the group not diagnosed with EoE is indicative of esophageal epithelial barrier disruption. Although EoE was diagnosed in only nine of these patients, this disruption may be an early risk marker for the development of esophagitis.

In 47 patients in whom EoE was not confirmed, immunohistochemical staining was positive for CCL-11 and negative for DSG-1. This might be explained by a subclinical course of the disease in this group of patients or that its signs may have been masked by drug use to treat allergy symptoms. It is a common phenomenon that patients swallow inhaled glucocorticoids, which may inadvertently “treat” them for EoE, preventing them from meeting the histopathological criteria for the diagnosis. However, most of the patients with histopathologically confirmed EoE were also taking corticosteroids. Another explanation for gastrointestinal symptoms in patients with positive CCL-11 staining may be oral allergy syndrome (OAS). This is supported by the fact that in this group, pollen allergy, with sensitivities to grass allergens (85%) and tree allergens (birch 66%, alder 68%, hazel 64%), predominated, in contrast to the subgroup with EoE (33%) and the subgroup with negative CCL-11 staining (13%). This correlation was statistically significant—*p* = 2.287 × 10^−7^ *(B). OAS is one of the manifestations of cross-reactive allergies and is characterized by clinical symptoms primarily localized in the oral cavity and throat, seen in patients with IgE-dependent allergic rhinitis after consuming certain foods. In this case, the clinical variant of OAS to consider is pollen food allergy syndrome (PFAS, PFS) [59,60,61].

We believe that this can be related to the occurrence of different phenotypes of EoE, with the “classic” form of the disease involving confirmed eosinophilia being the most extreme variant. In 2016, a new form (phenomenon) of eosinophilic esophagitis, referred to as EoE-like disease, was described by Straumann et al. The authors reported on a series of five patients with a family history of EoE presenting with symptoms typical for the disease, albeit with no esophageal eosinophilia. The main symptoms included severe dysphagia and chest pain (in two patients) while one patient had an episode of food bolus becoming entrapped in the esophagus. None of the patients improved after PPI treatment while rapid improvement was achieved following ingestible corticosteroids. This study laid the foundation for further research into the phenotypes of EoE [62]. In 2022, Greuter et al. reported on the characteristics of variants of eosinophilic esophagitis, as identified in their multicenter study. The study group consisted of 69 patients with different variants of EoE. Endoscopic abnormalities were detected in 53.6% of patients. A total of three histological subtypes were identified, including EoE-like esophagitis (36/69, 52.2%), lymphocytic esophagitis (14/69, 20.3%), and nonspecific esophagitis (19/69, 27.5%). In contrast to EoE, immunohistochemical assays showed no significant increase in inflammatory infiltration compared to GERD and the controls with the exception of lymphocytic infiltrates in lymphocytic esophagitis. The typical Th2 response in EoE was absent in all EoE variants. The authors confirmed that all the EoE variants were clinically and histologically active despite the absence of esophageal eosinophilia [6]. EoE variants appear to constitute a spectrum of diseases with classic EoE being the most common and prominent phenotype [7]. In 2023, Salvador Nunes et al. described different forms of EoE and distinguished between EoE-like esophagitis, lymphocytic esophagitis as a variant of EoE, non-specific esophagitis, and mast cell esophagitis. EoE-like esophagitis was defined as follows: the presence of 0–59 eos/mm^2^ (<15 eos/hpf) in esophageal biopsies with the presence of typical histopathological features of EoE, particularly dilated intercellular spaces and basal zone hyperplasia. Unlike the remaining two variants, EoE-like esophagitis presents with the highest risk of progressing to EoE over time. For this reason, regular follow-ups with endoscopy and esophageal biopsies are recommended. In addition, EoE-like esophagitis presents with the highest percentage of subepithelial eosinophils. Patients with EoE-like esophagitis also present with a higher incidence of fibrosis compared to patients with other variants of EoE. The pathogenesis of EoE-like esophagitis remains unclear, and appropriate diagnostic criteria also need to be established. Lymphocytic esophagitis is diagnosed on the basis of the following diagnostic criteria: the presence of inflammation with a predominance of lymphocytes, including large counts of intraepithelial lymphocytes (≥30/hpf) accumulated mainly in the peribronchiolar fields, dilatation of intercellular spaces, and the absence of intraepithelial granulocytes. In contrast to EoE, which is more common in younger males, lymphocytic esophagitis is more common in older women. Clinical presentation is similar to EoE, as dysphagia is the most common symptom. Endoscopic changes and treatment methods are also very similar to those observed in EoE. However, it is not yet clear whether lymphocytic esophagitis can be classified as a disease related to (i.e., a variant of) EoE or whether it is a separate and independent disease entity. Further studies are needed to provide a definite answer to this question. Nonspecific esophagitis is defined as histologically confirmed infiltration of lymphocytes or neutrophils that does not meet the diagnostic criteria for lymphocytic esophagitis [62,63,64]. To date, strict diagnostic criteria for this form of EoE have not been defined. Our study unequivocally demonstrated the existence of EoE variants without eosinophilia detectable in histopathological examination, with immunohistochemical assays for eotaxin-1 and desmoglein-1 potentially aiding in the diagnosis of these variants.

## 5. Conclusions

All patients with EoE who presented with multimorbidities were diagnosed with at least one allergic disease in addition to EoE. Positive staining for eotaxin-1 and negative staining for desmoglein-1, observed in patients with upper gastrointestinal symptoms and allergy who did not meet the diagnostic criteria for EoE, may suggest a subclinical course of the disease. This phenomenon could be attributed to the widespread use of corticosteroids among these patients. However, EoE likely manifests in different variant forms, with the “classic” form representing the most severe phenotype. Recognizing the potential presence of various phenotypes of eosinophilic esophagitis in patients with allergies and gastrointestinal symptoms may assist specialists in diagnosing and managing these patients more effectively.

## Figures and Tables

**Figure 1 nutrients-17-01355-f001:**
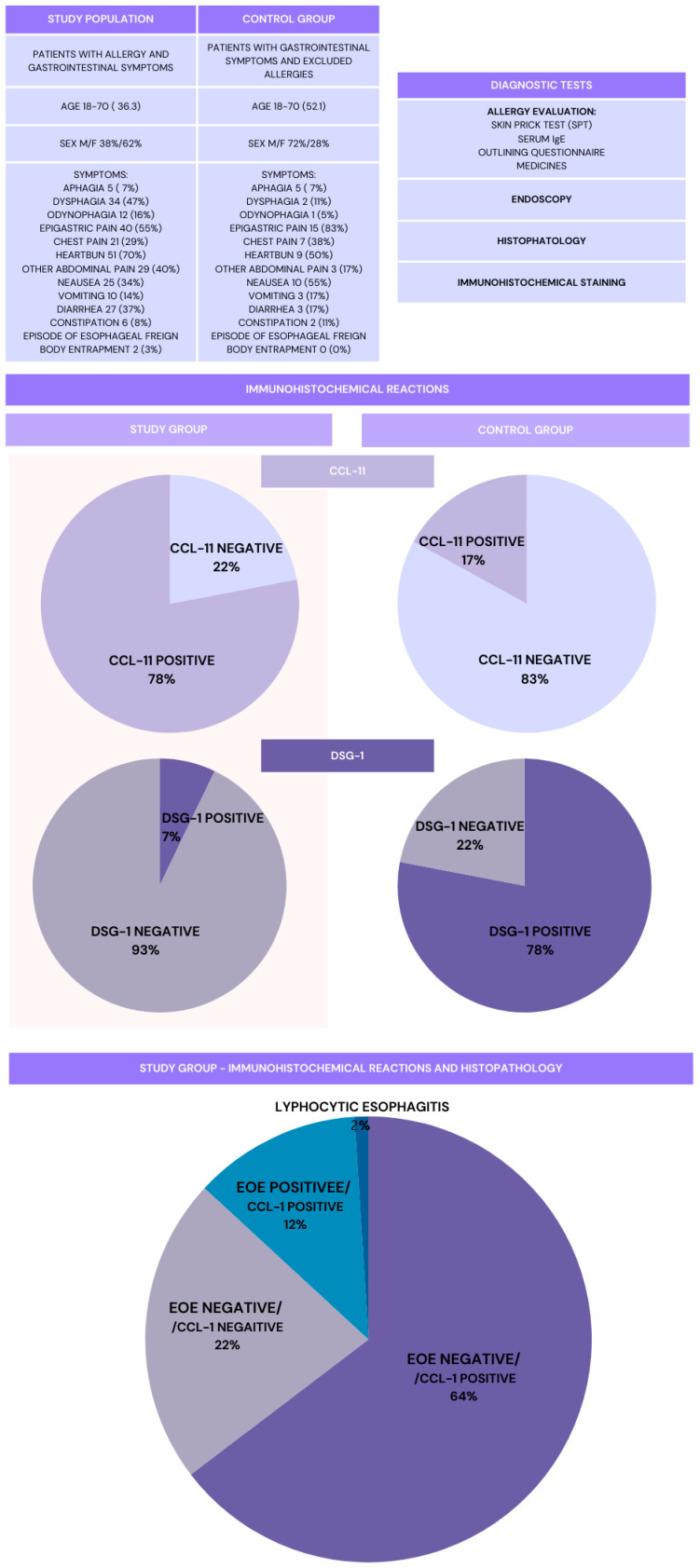
A schematic summary of the study.

**Table 1 nutrients-17-01355-t001:** Study group characteristics.

	Control Group *n* = 18	Allergy Group *n* = 73	*p*
Age, years, range (mean)	18–70 (52.1)	18–70 (36.3)	
Sex M/F, n (%)	13 (72)/5 (28)	28 (38)/45 (62)	0.2728/0.02023 *
Smoking n (%)	4 (22)	15 (20)	1
GI symptoms, n (%) Aphagia	2 (11)	5 (7)	1
Dysphagia	2 (11)	34 (47)	0.01289 *
Odynophagia	1 (5)	12 (16)	0.4204
Epigastric pain	15 (83)	40 (55)	0.05133
Chest pain	7 (39)	21 (29)	0.5835
Heatburn	9 (50)	51 (70)	0.1885
Other abdominal pain	3 (17)	29 (40)	0.1189
Nausea	10 (55)	25 (34)	0.1633
Vomiting	3 (17)	10 (14)	1
Diarrhea	3 (17)	27 (37)	0.173
Episode of esophageal foreign body entrapment	0	2 (3)	1
Esophagogastroscopy, n (%)			
Normal	18 (100)	61 (84)	0.145
Tracheolization	0	7 (9)	0.3823
Narrowing	0	2 (3)	1
White plaques	0	1 (1)	1
Erosions	0	3 (4)	0.8905
Rings	0	2 (3)	1
Histopathological findings, n (%)			
EOE positive (confirmed more than 15 eosinophils per HPF)	0	9 (12)	0.2591
Lymphocytic esophagitis	0	1 (1)	1
Immunohistochemical reactions, n (%)			
Eotaxin positive	3 (17)	56 (78)	6.7 × 10^−6^ * (B)
Eotaxin negative	15 (83)	17 (22)	6.7 × 10^−6^ * (B)
Desmoglein positive	14 (78)	6 (7)	1.317 × 10^−9^ * (B)
Desmoglein negative	4 (22)	67 (91)	1.317 × 10^−9^ * (B)
Drugs, n (%)			
INCS	0	48 (66)	
ICS	0	16 (22)	
Antihistamines	0	58 (79)	
PPI	4 (22)	14 (19)	1
SCS	0	1 (1)	1
INCS and ICS	0	14 (19)	0.0979
Allergy, n (%)			
AR	0	69 (94)	
Asthma	0	29 (27)	
AD	0	8 (10)	
FA	0	54 (73)	
Multimorbidity, n (%)			
AR + Asthma	0	19 (26)	
AR + FA	0	53 (72)	
AR + AD	0	8 (10)	
Asthma + FA	0	16 (21)	
Asthma + AD	0	4 (5)	
FA + AD	0	7 (9)	
AR + Asthma + FA	0	16 (21)	
AR + Asthma + AD	0	4 (5)	
AR + FA + AD	0	7 (9)	
Asthma + FA + AD	0	4 (5)	
AR + Asthma + FA + AD	0	4 (5)	

* statistically significant at the 0.05 level of significance; (B)—statistically significant after Bonferroni correction.

**Table 2 nutrients-17-01355-t002:** Allergy group characteristics according to histopathology and immunohistochemical results.

	EOE Positive Eotaxin Positive *n* = 9 (12%)	EoE Negative Eotaxin Positive *n* = 47 (64%)	EOE Negative Eotaxin Negative *n* = 16 (22%)	*p* Value
Immunohistochemical reactions, n (%)				
Desmoglein negative	9 (100)	47 (100)	11 (69)	8.248 × 10^−5^ * (B)
Desmoglein positive	0	0	5 (31)	8.248 × 10^−5^ * (B)
Glucocorticosteroids, n (%)				
INCS and ICS	6 (67)	8 (17)	0	0.0002192 * (B)
ICS	0	14 (30)	1 (6)	0.0348 *
SNS	0	1 (2)	0	0.7636
INCS	2 (22)	44 (94)	2 (12)	2.187 × 10^−10^
Endoscopy				
Normal, n (%)	2 (22)	42 (89)	16 (100)	6.069 × 10^−7^ * (B)
Trachealization, n (%)	6 (70)	0	0	1.124 × 10^−10^ * (B)
Narrowing, n (%)	2 (22)	0	0	0.0007466 *
White Plaques, n (%)	1 (11)	0	0	0.02874 *
Erosions, n (%)	0	3 (6)	0	0.4349
Rings, n (%)	0	2 (4)	0	0.5786
Allergy				
AR, n (%)	8 (89)	47 (100)	13 (76)	0.01355 *
Asthma, n (%)	6 (67)	10 (21)	3 (19)	0.01337 *
AD, n (%)	2 (22)	4 (9)	1 (6)	0.3866
FA, n (%)	7 (78)	30 (64)	16 (100)	0.01715 *
Allergens skin prick tests, n (%)				
Dermatophagoides farinae	8 (89)	25 (53)	4 (25)	0.00828 *
Dermatophagoides pterynossinus	8 (89)	26 (55)	4 (25)	0.007505 *
Grasses (mix)	3 (33)	40 (85)	2 (13)	2.287 × 10^−7^ * (B)
Alternaria	1 (11)	5 (10)	4 (25)	0.3456
Aspergillus	0	7 (15)	2 (13)	0.4649
Cat	1 (11)	19 (40)	6 (38)	0.2428
Dog	0	12 (26)	4 (25)	0.2298
Alder	3 (33)	32 (68)	6 (38)	0.03186 *
Birch	8 (89)	31 (66)	8 (50)	0.1444
Hazel Tree	3 (33)	30 (64)	9 (56)	0.2314
Rye	3 (33)	34 (72)	4 (25)	0.001327 *
Common Mugwort	1 (11)	16 (34)	7 (44)	0.2476
Ribwort Plantain	0	13 (28)	7 (44)	0.06404
Milk	5 (56)	5 (10)	0	0.0003255 *
Hen’s Egg	1 (11)	4 (9)	2 (13)	0.8873
Wheat	7 (78)	5 (10)	1 (6)	3.819 × 10^−6^ * (B)
Hazelnuts	9 (100)	3 (6)	9 (56)	2.857 × 10^−9^ * (B)
Soybeans	1 (11)	0	1 (6)	0.1124
Cod	1 (11)	2 (4)	1 (6)	0.7063
Allergens sIgE, n (%)				
Dermatophagoides farinae	8 (89)	10 (21)	4 (25)	0.000252 *
Dermatophagoides pteronyssinus	8 (89)	10 (21)	3 (19)	0.0001368 *
Alternaria	1 (11)	4 (9)	0	0.4461
Cat	1 (11)	10 (21)	7 (44)	0.118
Dog	0	10 (21)	1 (6)	0.1397
Alder	3 (33)	29 (62)	2 (13)	0.002039 *
Birch	8 (89)	30 (64)	4 (25)	0.003421 *
Hazel Tree	3 (33)	25 (53)	3 (19)	0.04569 *
Grasses (Mix)	3 (33)	25 (53)	4 (25)	0.1132
Rye	3 (33)	20 (43)	5 (31)	0.6787
Common Mugwort	1 (11)	10 (21)	4 (25)	0.7083
Ribwort Plantain	0	3 (6)	4 (25)	0.05444
Milk	2 (22)	4 (9)	2 (13)	0.4776
Hen’s Egg	1 (11)	4 (9)	0	0.4461
Wheat	5 (56)	5 (10)	2 (13)	0.003643 *
Hazelnuts	6 (67)	2 (4)	0	9.392 × 10^−8^ * (B)
Soybeans	1 (11%)	0	1 (6)	0.1124
Cod	2 (22)	3 (6)	1 (6)	0.2728

* statistically significant at the 0.05 level of significance; (B)—statistically significant after Bonferroni correction.

## Data Availability

Data presented in this study are available on request from the corresponding author.

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
