# Peer review of "The Overlap of Allergic Disorders and Upper Gastrointestinal Symptoms: Beyond Eosinophilic Esophagitis"

_nutrients, 2025, doi:10.3390/nu17081355_

Round 1
Reviewer 1 Report
Comments and Suggestions for Authors
- Significance of the Study
This study is clinically significant as it investigates the prevalence of EoE among patients with allergies and highlights the challenges in diagnosing the disease. In particular, the finding that abnormal inflammatory markers were observed even in patients who were not diagnosed with EoE is intriguing, as it suggests the possibility of a novel disease concept.
However, additional supporting data are required to strengthen the findings of this study.
Biomarkers for EoE Diagnosis
-Eotaxin-1 (CCL-11) and desmoglein-1 (DSG-1) were used as biomarkers for EoE diagnosis. What are their sensitivity and specificity in this context?
-Additionally, what was the rationale for selecting patients with functional dyspepsia (FD) and irritable bowel syndrome (IBS) without allergies as the control group?
Author Response
Dear Sir/Madame,
Thank you very much for your kind and encouraging feedback. We sincerely appreciate your thoughtful review and the time you dedicated to helping us improve our manuscript.
-Eotaxin-1 (CCL-11) and desmoglein-1 (DSG-1) were used as biomarkers for EoE diagnosis. What are their sensitivity and specificity in this context?
REPLY:
Assessing the sensitivity and specificity of CCL-11 and DSG-1 biomarkers in the diagnosis of eosinophilic esophagitis (EoE) remains challenging, as they are not yet included in the established diagnostic criteria for this condition Unfortunately, neither sensitivity nor specificity have yet been reliably estimated in the available literature at present.
-Additionally, what was the rationale for selecting patients with functional dyspepsia (FD) and irritable bowel syndrome (IBS) without allergies as the control group?
REPLY:
We selected such control group to avoid patients with GERD and other diseases that are related with esophageal eosinophilic infiltration. We could not recruit healthy controls as gastroscopy in such population would not be justified.

Reviewer 2 Report
Comments and Suggestions for Authors
This manuscript explores the relationship between upper gastrointestinal (GI) symptoms, allergy, and eosinophilic esophagitis (EoE), focusing on the prevalence of EoE among patients with food and inhalant allergies. The authors present histopathological and immunohistochemical findings, particularly eotaxin-1 (CCL-11) and desmoglein-1 (DSG-1) staining, to investigate potential subclinical cases of EoE. The study is clinically relevant, given the overlap between allergic disorders and eosinophilic GI diseases, and it provides useful data on multimorbidity in allergic patients. However, several key areas need improvement, particularly the study’s design clarity, the interpretation of findings, and the discussion of potential confounding factors. For these reasons, I recommend major revisions before acceptance.
Major Concerns and Areas for Improvement
- The inclusion and exclusion criteria are not sufficiently detailed, particularly regarding the diagnostic workup for gastroesophageal reflux disease (GERD) and other non-EoE causes of esophageal eosinophilia. Did all patients undergo pH monitoring or empirical PPI therapy before being classified as non-GERD? Were patients with functional dyspepsia or achalasia systematically excluded? Given the high prevalence of GERD and its potential overlap with EoE, these factors must be explicitly addressed in the methodology.
- The study suggests that patients with upper GI symptoms, allergy, and positive eotaxin-1 staining (but without histological EoE criteria) may have a subclinical form of the disease.
- However, without functional data on epithelial barrier integrity or inflammatory pathways, this conclusion is speculative.
- Were patients with non-EoE esophageal eosinophilia tested for other biomarkers of chronic inflammation (e.g., IL-5, TGF-β)?
- The negative DSG-1 staining is intriguing, but its clinical significance remains unclear without a mechanistic explanation. The discussion should include potential alternative explanations for DSG-1 loss, including nonspecific epithelial injury or prior corticosteroid use.
- The authors correctly acknowledge the potential masking effect of inhaled corticosteroids (ICS) and intranasal corticosteroids (InCS) on EoE. However, they do not provide a stratified analysis comparing patients who were and were not using these medications at the time of biopsy. A sub-analysis excluding patients on corticosteroids would help clarify whether corticosteroid exposure influenced the histopathological findings.
- The study reports statistically significant differences between groups, but does not adjust for potential confounders such as age, sex, atopic history, and corticosteroid use. A multivariate regression analysis should be performed to determine independent predictors of positive eotaxin-1 staining and EoE diagnosis. This would help determine whether allergy severity, specific allergens, or medication use influenced the results.
- The study focuses primarily on allergy and immune factors but does not explore the potential role of the esophageal and gut microbiome in EoE. Recent research suggests that microbial dysbiosis may contribute to EoE pathogenesis, particularly through epithelial barrier dysfunction and immune modulation. The discussion should include a broader perspective on potential environmental and microbiome-related factors contributing to esophageal inflammation.
To further strengthen the manuscript, the authors should cite and integrate PMID: 38667503 in the discussion. PMID: 38667503 explores the role of epithelial barrier dysfunction in EoE and its relationship with allergic inflammation. The study's findings on desmoglein-1 loss could be better contextualized by referencing PMID: 38667503, which discusses how barrier dysfunction may drive eosinophilic inflammation even in the absence of overt eosinophilia. The authors should highlight the potential clinical implications of barrier dysfunction in at-risk allergic patients, citing this reference to support the hypothesis that EoE represents a spectrum of disease rather than a binary diagnosis.
Author Response
Responses to the review:
Thank you very much for your time and valuable comments included the review.
- The inclusion and exclusion criteria are not sufficiently detailed, particularly regarding the diagnostic workup for gastroesophageal reflux disease (GERD) and other non-EoE causes of esophageal eosinophilia. Did all patients undergo pH monitoring or empirical PPI therapy before being classified as non-GERD? Were patients with functional dyspepsia or achalasia systematically excluded? Given the high prevalence of GERD and its potential overlap with EoE, these factors must be explicitly addressed in the methodology.
We corrected the description of inclusion criteria. Nether manometry, nor pH monitoring nor impedance were included in the study protocol, however all patients were carefully examined including detailed history analysis and gastroscopy moreover the majority of study participants had pH monitoring or manometry performed outside our centre.
- The study suggests that patients with upper GI symptoms, allergy, and positive eotaxin-1 staining (but without histological EoE criteria) may have a subclinical form of the disease.
Indeed, it seems to us that in the group of patients with upper gastrointestinal symptoms, allergy and positive staining for CCL-11 and negative staining for DSG-1, a subclinical form of EoE or one of the phenotypic forms of EoE may occur without meeting the criteria of 15 eosinophils in the field of view in the histopathological examination.
- However, without functional data on epithelial barrier integrity or inflammatory pathways, this conclusion is speculative.
It is speculative- we agree. However current methods to assess integrity of inflammatory pathways are also indirect. We have assessed the morphology ( including inflammatory infiltration) in each patient using standard HE staining of oesophageal samples.
- Were patients with non-EoE esophageal eosinophilia tested for other biomarkers of chronic inflammation (e.g., IL-5, TGF-β)?
Indeed, a number of proinflammatory cytokines and chemokines play an important role in the pathogenesis of EoE. Unfortunately, in our study we did not examine IL-5 and TGF-beta, but this will certainly be the subject of our further studies.
- The negative DSG-1 staining is intriguing, but its clinical significance remains unclear without a mechanistic explanation. The discussion should include potential alternative explanations for DSG-1 loss, including nonspecific epithelial injury or prior corticosteroid use.
We have amended the discussion accordingly.
- The authors correctly acknowledge the potential masking effect of inhaled corticosteroids (ICS) and intranasal corticosteroids (InCS) on EoE. However, they do not provide a stratified analysis comparing patients who were and were not using these medications at the time of biopsy. A sub-analysis excluding patients on corticosteroids would help clarify whether corticosteroid exposure influenced the histopathological findings.
Tables no. 2 have been supplemented with the glucocorticosteroids taken during biopsy. All patients except one from the EoE group were taking steroids. In the group with unconfirmed EoE but positive staining for CCL-11, the vast majority of patientswere taking steroids. In the group with negative staining for CCL-11, very few patients took steroids, which indicates a mild course of their allergic disease.
………………………………………………………………………………………………….
- The study reports statistically significant differences between groups, but does not adjust for potential confounders such as age, sex, atopic history, and corticosteroid use. A multivariate regression analysis should be performed to determine independent predictors of positive eotaxin-1 staining and EoE diagnosis. This would help determine whether allergy severity, specific allergens, or medication use influenced the results.
Thank you, this is a valuable comment. We performed a logistic multiple regression model in two variants, when the response variable was EoE diagnosis and positive eotaxin-1 staining adjusted for the age, sex, type of allergy, DSG, ICS, INCS. However, sex, age and type of allergy were not
significant, which confirms that the results of previous statistical analyses are not biased by the influence of these confounders
- The study focuses primarily on allergy and immune factors but does not explore the potential role of the esophageal and gut microbiome in EoE. Recent research suggests that microbial dysbiosis may contribute to EoE pathogenesis, particularly through epithelial barrier dysfunction and immune modulation. The discussion should include a broader perspective on potential environmental and microbiome-related factors contributing to esophageal inflammation.
The pathogenesis of eosinophilic esophagitis (EoE) is undoubtedly multifactorial and involves the interplay of genetic predisposition, environmental factors, and dysregulated immune responses. The immune dysregulation in EoE is predominantly Th2-driven and is primarily mediated by cytokines IL-4, IL-5, and IL-13, which are responsible for eosinophil recruitment [1,2].
Another key aspect of EoE pathogenesis is epithelial barrier dysfunction, which facilitates the penetration of allergens and perpetuates the inflammatory response. Numerous environmental factors, including diet, cesarean delivery, childhood antibiotic use, formula feeding, early exposure to proton pump inhibitors (PPIs), cold climate, and poor living conditions, have been associated with an increased risk of EoE [3]. Conversely, some studies suggest that childhood exposure to furry pets and Helicobacter pylori infection may be linked to a reduced risk of developing EoE [3,4].
The role of the gut microbiota in immune regulation is increasingly recognized as significant [5]. Therapeutic strategies aimed at modulating microbiota composition to alleviate inflammation and allergen sensitization may hold promise in EoE treatment [5,6,7,8,9,10].
A study by Zhang et al. found an increased abundance of Haemophilus and Aggregatibacter species, along with a decrease in Firmicutes, in patients with active EoE [11]. Meanwhile, research by Arias et al. demonstrated that early interactions between epithelial cells and the esophageal microbiota could modulate CXCL16 expression and recruit invariant natural killer T (iNKT) cells to the esophageal epithelium [12]. In a mouse model, administration of Lactococcus lactis NCC 2287 induced histopathologic remission of EoE [13].
The discussion has been amended accordingly.
- Massironi S, Mulinacci G, Gallo C, Elvevi A, Danese S, Invernizzi P, Vespa E. Mechanistic Insights into Eosinophilic Esophagitis: Therapies Targeting Pathophysiological Mechanisms. Cells. 2023 Oct 18;12(20):2473. doi: 10.3390/cells12202473. PMID: 37887317; PMCID: PMC10605530.
- Barchi A, Vespa E, Passaretti S, Dell'Anna G, Fasulo E, Yacoub MR, Albarello L, Sinagra E, Massimino L, Ungaro F, Danese S, Mandarino FV. The Dual Lens of Endoscopy and Histology in the Diagnosis and Management of Eosinophilic Gastrointestinal Disorders-A Comprehensive Review. Diagnostics (Basel). 2024 Apr 22;14(8):858. doi: 10.3390/diagnostics14080858. PMID: 38667503; PMCID: PMC11049211.
- Lucendo, A.J.; Santander, C.; Savarino, E.; Guagnozzi, D.; Pérez-Martínez, I.; Perelló, A.; Guardiola-Arévalo, A.; Barrio, J.; Elena Betoré-Glaria, M.; Gutiérrez-Junquera, C.; et al. EoE CONNECT, the European Registry of Clinical, Environmental, and Genetic Determinants in Eosinophilic Esophagitis: Rationale, Design, and Study Protocol of a Large-Scale Epidemiological Study in Europe. Ther. Adv. Gastroenterol. 2022, 15, 17562848221074204
- Biedermann, L.; Straumann, A.; Greuter, T.; Schreiner, P. Eosinophilic Esophagitis-Established Facts and New Horizons. Semin. Immunopathol. 2021, 43, 319–335
- Massimino, L.; Barchi, A.; Mandarino, F.V.; Spanò, S.; Lamparelli, L.A.; Vespa, E.; Passaretti, S.; Peyrin-Biroulet, L.; Savarino, E.V.; Jairath, V.;
et al. A Multi-Omic Analysis Reveals the Esophageal Dysbiosis as the Predominant Trait of Eosinophilic Esophagitis. J. Transl. Med. 2023, 21, 46.
- Angerami Almeida, K.; de Queiroz Andrade, E.; Burns, G.; Hoedt, E.C.; Mattes, J.; Keely, S.; Collison, A. The Microbiota in Eosinophilic Esophagitis: A Systematic Review. J. Gastroenterol. Hepatol. 2022, 37, 1673–1684
- Benitez, A.J.; Hoffmann, C.; Muir, A.B.; Dods, K.K.; Spergel, J.M.; Bushman, F.D.; Wang, M.-L. Inflammation-Associated Microbiota in Pediatric Eosinophilic Esophagitis. Microbiome 2015, 3, 23
- Harris, J.K.; Fang, R.; Wagner, B.D.; Choe, H.N.; Kelly, C.J.; Schroeder, S.; Moore, W.; Stevens, M.J.; Yeckes, A.; Amsden, K.; et al. Esophageal Microbiome in Eosinophilic Esophagitis. PLoS ONE 2015, 10, e0128346
- Laserna-Mendieta, E.J.; FitzGerald, J.A.; Arias-Gonzalez, L.; Ollala, J.M.; Bernardo, D.; Claesson, M.J.; Lucendo, A.J. Esophageal Microbiome in Active Eosinophilic Esophagitis and Changes Induced by Different Therapies. Sci. Rep. 2021, 11, 7113
- Furuta, G.T.; Fillon, S.A.; Williamson, K.M.; Robertson, C.E.; Stevens, M.J.; Aceves, S.S.; Arva, N.C.; Chehade, M.; Collins, M.H.; Davis, C.M.; et al. Mucosal Microbiota Associated with Eosinophilic Esophagitis and Eosinophilic Gastritis. J. Pediatr. Gastroenterol. Nutr. 2023, 76, 347–354
- Zhang, X.; Zhang, N.; Wang, Z. Eosinophilic Esophagitis and Esophageal Microbiota. Front. Cell Infect. Microbiol. 2023, 13, 1206343
- Arias, Á.; Lucendo, A.J. Molecular Basis and Cellular Mechanisms of Eosinophilic Esophagitis for the Clinical Practice. Expert. Rev. Gastroenterol. Hepatol. 2019, 13, 99–117
- Karpathiou, G.; Papoudou-Bai, A.; Ferrand, E.; Dumollard, J.M.; Peoc’h, M. STAT6: A Review of a Signaling Pathway Implicated in Various Diseases with a Special Emphasis in Its Usefulness in Pathology. Pathol. Res. Pract. 2021, 223, 153477.
- To further strengthen the manuscript, the authors should cite and integrate PMID: 38667503 in the discussion. PMID: 38667503 explores the role of epithelial barrier dysfunction in EoE and its relationship with allergic inflammation. The study's findings on desmoglein-1 loss could be better
contextualized by referencing PMID: 38667503, which discusses how barrier dysfunction may drive eosinophilic inflammation even in the absence of overt eosinophilia. The authors should highlight the potential clinical implications of barrier dysfunction in at-risk allergic patients, citing this reference
to support the hypothesis that EoE represents a spectrum of disease rather than a binary diagnosis.
We sincerely appreciate the recommendation of this highly intriguing article.
Thank you very much for this comment . We have cited aforementioned studies.

Reviewer 3 Report
Comments and Suggestions for Authors
Polish authors investigated symptoms from the upper gastrointestinal tract in patients with allergy in relation with eosinophilic esophagitis(EoE). My concerns are as follows.
1. The aim described in Abstract was to evaluate the prevalence of EoE in a group of patients with upper gastrointestinal symptoms and food and/or inhalant allergies, and that in the main text was to evaluate the esophageal mucosa in a group of patients with inhalant and food allergies and esophageal symptoms. The authors' conclusion was “Positive staining for eotaxin-1 and negative staining for desmoglein-1 in patients with esophageal symptoms and allergy but who did not met EoE diagnostic criteria, could be indicative of subclinical course of the disease or masking effect of corticosteroids”. Considering their conclusion, the aim of this prospective study is better to be determined more specifically and clearly.
2. In spite of redundant long discussion, the authors’ hypothesis that non-EoE patients with positive staining for eotaxin-1 and negative staining for desmoglein-1 could be indicative of subclinical course of the disease or masking effect of corticosteroids was neither adequately validated nor proven in this manuscript. The three groups shown in Table 2 were defined only with eotaxin positive or negative. I wonder how was desmoglein in the three groups. I would like to recommend the authors to clarify clinical and pathophysiological characteristics of this patient group with allergy and non-EoE compared with EoE and non-EoE/non-allergy patient groups.
3. Figure 1 must be upgraded to be made out easily by readers.
4. There are some typos and grammatical errors in this manuscript. I strongly recommend the authors to receive professional editing in terms of language and composition.
noted above
Author Response
Thank you very much for your time and valuable comments in the review
responses to the reviews:
- The aim described in Abstract was to evaluate the prevalence of EoE in a group of patients with upper gastrointestinal symptoms and food and/or inhalant allergies, and that in the main text was to evaluate the esophageal mucosa in a group of patients with inhalant and food allergies and esophageal symptoms. The authors' conclusion was “Positive staining for eotaxin-1 and negative staining for desmoglein-1 in patients with esophageal symptoms and allergy but who did not met EoE diagnostic criteria, could be indicative of subclinical course of the disease or masking effect of corticosteroids”. Considering their conclusion, the aim of this prospective study is better to be determined more specifically and clearly.
The objective of this study was to evaluate the esophageal mucosa in a group of patients with inhalant and food allergies and esophageal symptoms. We have not expected obtained results, this is an unexpected find and is merely speculation/suspicion, not a hard conclusion
- In spite of redundant long discussion, the authors’ hypothesis that non-EoE patients with positive staining for eotaxin-1 and negative staining for desmoglein-1 could be indicative of subclinical course of the disease or masking effect of corticosteroids was neither adequately validated nor proven in this manuscript. The three groups shown in Table 2 were defined only with eotaxin positive or negative. I wonder how was desmoglein in the three groups. I would like to recommend the authors to clarify clinical and pathophysiological characteristics of this patient group with allergy and non-EoE compared with EoE and non-EoE/non-allergy patient groups.
We have made an effort to restructure the discussion. In all patients from the group diagnosed with EoE (based on histopathological criteria) and with positive staining for CCL-11, negative staining for DSG-1 was observed. In the group of patients without a confirmed EoE diagnosis but with positive staining for CCL-11, negative staining for DSG-1 was also found. Conversely, in the group of patients without a confirmed EoE diagnosis and with negative staining for CCL-11, 11 patients exhibited negative staining for DSG-1, while 6 showed positive staining for DSG-1.
Assuming that positive DSG-1 staining indicates an intact epithelial barrier, it can be inferred that the epithelial barrier was compromised in all patients with EoE, all patients with an "EoE-like" condition, and a subset of patients with negative CCL-11 staining. In the control group, positive staining for CCL-11 was observed in 3 patients, whereas positive DSG-1 staining was found in 14 patients. This suggests that, in the majority of patients within the control group, the epithelial barrier remained intact. We acknowledge that this is a simplification; however, a more in-depth exploration of this topic requires further research. These findings should be considered preliminary. We have incorporated this data into the table.
- Figure 1 must be upgraded to be made out easily by readers
We tried to improve the figure
……………………………………………………………………………
- There are some typos and grammatical errors in this manuscript. I strongly recommend the authors to receive professional editing in terms of language and composition
We have reviewed the text.

Reviewer 4 Report
Comments and Suggestions for Authors
The manuscript addresses an important clinical question regarding the prevalence of eosinophilic esophagitis (EoE) in patients with allergies and upper gastrointestinal symptoms. The study is well-structured and presents a novel approach by screening allergic patients for EoE. However, several aspects need improvement
Major Comments:
- The title could be more specific to reflect the core findings of the study. A possible revision: "The Overlap of Allergic Disorders and Upper Gastrointestinal Symptoms: Beyond Eosinophilic Esophagitis?"
- The abstract lacks specificity in the conclusion. It should clearly highlight the key findings and their implications for clinical practice.
- The introduction should provide a clearer rationale for why investigating EoE in allergic patients is crucial. Some statements require citations, particularly regarding the prevalence of EoE in different populations, and the mechanistic insights. The discussion about different EoE phenotypes is relevant but should be condensed for brevity.
- The demographic differences between study and control groups are notable, particularly the significant age disparity (mean age 36.3 vs. 52.1 years). Could age differences influence findings? Please add some citations.
- The study concludes that EoE is rare, yet 12% prevalence in this cohort is quite high. This should be framed appropriately.
- The discussion on positive staining for eotaxin-1 (CCL-11) and absence of desmoglein-1 (DSG-1) is intriguing but requires a more direct explanation regarding its diagnostic utility.
- The manuscript suggests that corticosteroid use may have masked EoE features, leading to "subclinical disease." However, the authors should clarify how they accounted for this confounder.
- The concept of "EoE-like disease" is interesting but should be critically appraised with supporting evidence.
- The findings should be discussed in the context of existing literature, particularly the role of environmental exposures and genetic predisposition.
- The conclusion should explicitly state the clinical implications of the findings.
- The potential for immunohistochemical markers (CCL-11, DSG-1) to serve as early diagnostic indicators should be cautiously interpreted.
Minor Comments:
- There are multiple typographical errors throughout the text, e.g., "Symtpoms" should be "Symptoms." A thorough proofread is necessary.
- Abbreviations such as "ICS" and "InCS" should be defined at first mention.
- The reference list is extensive but contains inconsistencies in formatting. Ensure uniform citation style.
- Figures and tables should be labeled clearly and referenced in the text appropriately.
- Some sentences are overly long and difficult to follow. Consider breaking them down for better readability.
There are multiple typographical errors throughout the text. A thorough proofread is necessary.
Author Response
Dear Sir/Madame,
Thank you very much for your feedback. We sincerely appreciate your thoughtful review and the time you dedicated to helping us improve our manuscript.
Major comments:
- The title could be more specific to reflect the core findings of the study. A possible revision: "The Overlap of Allergic Disorders and Upper Gastrointestinal Symptoms: Beyond Eosinophilic Esophagitis?"
We will change the title according to your suggestion.
- The abstract lacks specificity in the conclusion. It should clearly highlight the key findings and their implications for clinical practice.
We have completed the summary according to your suggestion:
Conclusions. Eoe is rare disease usually accompanied by allergic multimorbidity. Positive staining for eotaxin-1 and negative staining for desmoglein-1 in patients with esophageal symptoms and allergy but who did not met EoE diagnostic criteria, could be indicative of subclinical course of the disease or masking effect of corticosteroids. It is now critically important for both researchers and practicing clinicians to recognize that eosinophilic esophagitis (EoE) is not a homogeneous disease but rather consists of multiple subtypes (phenotypes). The so-called "classic" form of EoE—defined by current diagnostic criteria as having more than 15 eosinophils per high-power field on histopathological examination—appears to represent only the tip of the iceberg. There is an urgent need for further research to refine endoscopic techniques, expand the scope of histopathological assessments, and identify novel biomarkers to better define the distinct phenotypes of eosinophilic esophagitis.
- The introduction should provide a clearer rationale for why investigating EoE in allergic patients is crucial. Some statements require citations, particularly regarding the prevalence of EoE in different populations, and the mechanistic insights. The discussion about different EoE phenotypes is relevant but should be condensed for brevity.
We have added a supplement to the introduction in accordance with your suggestion:
Eosinophilic esophagitis (EoE) is a chronic immune-mediated inflammatory disease characterized by esophageal dysfunction resulting from an abnormal Th2-type immune response. Dysregulated immune activation triggered by exposure to airborne and food allergens plays a crucial role in the pathogenesis of this condition. This leads to inflammation and tissue damage in the esophagus, causing the characteristic symptoms and histopathological changes observed in patients with EoE. Key elements in the pathogenesis of EoE include allergen sensitization, Th2-driven immune response, epithelial barrier dysfunction, eosinophilic infiltration, fibrosis, and tissue remodeling. Additionally, genetic and environmental factors also play a significant role in disease development.
We believe we have cited all relevant data regarding the prevalence of EoE in different populations in the discussion. We have also shortened this description in the discussion section.
- The demographic differences between study and control groups are notable, particularly the significant age disparity (mean age 36.3 vs. 52.1 years). Could age differences influence findings? Please add some citations.
We performed a logistic regression model in two variants, when the response variable was EoE diagnosis and positive eotaxin-1 staining adjusted for the age. However, age was not significant, which confirms that the results of previous statistical analyses are not biased by the influence of age. Eosinophilic esophagitis is a disease that occurs most often in the 3rd-5th decade of life, although also in the pediatric population. Allergy is also a disease that occurs most often in young people. That is probably why the average age of the subjects with allergy and upper gastrointestinal symptoms was lower than in patients without allergy and with upper gastrointestinal symptoms.
[Steinbach EC, Hernandez M, Dellon ES. Eosinophilic Esophagitis and the Eosinophilic Gastrointestinal Diseases: Approach to Diagnosis and Management. J Allergy Clin Immunol Pract. 2018 Sep- Oct;6(5):1483-1495. doi: 10.1016/j.jaip.2018.06.012. Epub 2018 Jul 3. PMID: 30201096; PMCID: PMC6134874.]
- The study concludes that EoE is rare, yet 12% prevalence in this cohort is quite high. This should be framed appropriately
We have added the following entry:
In our study, among 73 individuals with allergies and upper gastrointestinal symptoms, eosinophilic esophagitis (EoE) was diagnosed in 12% of cases. On one hand, a higher prevalence might have been expected; however, within our specific cohort, this represents a relatively significant finding.
- The discussion on positive staining for eotaxin-1 (CCL-11) and absence of desmoglein-1 (DSG-1) is intriguing but requires a more direct explanation regarding its diagnostic utility.
We selected the DSG-1 biomarker based on the fact that desmoglein-1 belongs to the cadherin family, which plays a crucial role in the formation of desmosomes—structures responsible for intercellular adhesion in the epithelium. Sherrill et al. demonstrated that in patients with eosinophilic esophagitis (EoE), there is a reduction in desmoglein-1 expression in the esophageal epithelium, which disrupts epithelial integrity and weakens the epithelial barrier. Moreover, the authors suggest that desmoglein-1 downregulation is associated with the pro-inflammatory cytokine IL-13 [Sherrill, JD.; Kc, K.; Wu, D.; Djukic, Z.; Caldwell, JM.; Stucke, EM.; et al. Desmoglein-1 regulates esophageal epithelial barrier function and immune responses in eosinophilic esophagitis. Mucosal Immunol. 2014;7(3):718-29. doi: 10.1038/mi.2013.90]. The observed decrease in desmoglein-1 levels in our patients may therefore indicate impaired esophageal epithelial barrier function.
In 2025, Hoelz et al. [Hoelz H, Faro T, Frank ML, Forné I, Kugelmann D, Jurk A, Buehler S, Siebert K, Matchado M, Straub T, Hering A, Piontek G, Mueller S, Koletzko S, List M, Steiger K, Rudelius M, Waschke J, Schwerd T. Persistent desmoglein-1 downregulation and periostin accumulation in histologic remission of eosinophilic esophagitis. J Allergy Clin Immunol. 2025 Feb;155(2):505-519. doi: 10.1016/j.jaci.2024.09.016. Epub 2024 Sep 27. PMID: 39343172.] characterized molecular alterations in pediatric patients with EoE at different disease stages using tissue transcriptomics and proteomics. Their findings revealed that even in clinical and histopathological remission, desmosome regeneration was impaired, and desmoglein-1 expression remained reduced.
Eotaxin-1 (CCL11) is a chemokine with strong chemotactic and activating effects on eosinophils. It plays a key role in eosinophil recruitment to the bronchial mucosa in patients with asthma and is likely to have a similar function in eosinophilic esophagitis. Eosinophils infiltrating the esophageal mucosa produce mediators responsible for tissue damage, leading to mucosal remodeling. Recent studies have also shown that eotaxin-1 is involved in aging processes. The presence of eotaxin-1 in the esophageal mucosa of patients with EoE reflects ongoing eosinophilic inflammation. Furthermore, positive staining for eotaxin-1 in the mucosa of patients with allergies and gastrointestinal symptoms, but without confirmed EoE, may suggest a subclinical form of the disease or an increased predisposition of the mucosa to eosinophilic inflammation.
[Fan, Z.; Kong, M.; Dong, W.; Dong, C.; Miao, X.; Guo, Y.; et al. Trans-activation of eotaxin-1 by Brg1 contributes to liver regeneration. Cell Death Dis. 2022;13(5):495. doi: 10.1038/s41419-022-04944-0; Lavandoski, P.; Pierdoná, V.; Maurmann, RM.; Grun, LK.; Guma, FTCR.; et al. Eotaxin-1/CCL11 promotes cellular senescence in human-derived fibroblasts through pro-oxidant and pro-inflammatory pathways. Front Immunol. 2023;14:1243537. doi: 10.3389/fimmu.2023.1243537; Nino, G.; Huseni, S.; Perez, GF.; Pancham, K.; Mubeen, H.; Abbasi, A.; et al. Directional secretory response of double-stranded RNA-induced thymic stromal lymphopoietin (TSLP) and CCL11/eotaxin-1 in human asthmatic airways. PLoS One. 2014;9(12):e115398. doi: 10.1371/journal.pone.0115398; Lucendo, AJ.; De Rezende, L.; Comas, C.; Caballero, T.; Bellón, T. Treatment with topical steroids downregulates IL-5, eotaxin-1/CCL11, and eotaxin-3/CCL26 gene expression in eosinophilic esophagitis. Am J Gastroenterol. 2008;103(9):2184-93. doi: 10.1111/j.1572-0241.2008.01937.x].
- The manuscript suggests that corticosteroid use may have masked EoE features, leading to "subclinical disease." However, the authors should clarify how they accounted for this confounder.
…Tables no. 2 have been supplemented with the glucocorticosteroids taken during biopsy. All patients except one from the EoE group were taking steroids. In the group with unconfirmed EoE but positive staining for CCL-11, the vast majority of patientswere taking steroids. In the group with negative staining for CCL-11, very few patients took steroids, which indicates a mild course of their allergic disease……………………………………………………………………………………………………………
- The concept of "EoE-like disease" is interesting but should be critically appraised with supporting evidence.
The concept of "EoE-like disease" was taken from the work :
- Greuter, T.; Straumann, A.; Fernandez-Marrero, Y.; Germic, N.; Hosseini, A.; Yousefi, S.; et al. Characterization of eosinophilic esophagitis variants by clinical, histological, and molecular analyses: A cross-sectional multi-center study.
Allergy. 2022;77(8):2520-2533. doi: 10.1111/all.15233. ;
- Greuter, T.; Straumann, A.; Fernandez-Marrero, Y.; Germic, N.; Hosseini, A.; Chanwangpong, A.; et al.
A Multicenter Long-Term Cohort Study of Eosinophilic Esophagitis Variants and Their Progression to Eosinophilic Esophagitis Over Time.
Clin Transl Gastroenterol. 2024 1;15(4):e00664. Doi: 10.14309/ctg.0000000000000664.
- Salvador Nunes, VS.; Straumann, A.; Salvador Nunes, L.; Schoepfer, AM.; Greuter, T.
Eosinophilic Esophagitis beyond Eosinophils - an Emerging Phenomenon Overlapping with Eosinophilic Esophagitis: Collegium Internationale Allergologicum (CIA) Update 2023. Int Arch Allergy Immunol. 2023;184(5):411-420. doi: 10.1159/000529910
In 2022, Greuter et al. reported on the characteristics of variants of eosinophilic esophagitis as identified in their multicenter study. The study group consisted of 69 patients with different variants of EoE. Endoscopic abnormalities were detected in 53.6% of patients. A total of three histological subtypes were identified, including EoE-like esophagitis (36/69, 52.2%), lymphocytic esophagitis (14/69, 20.3%), and nonspecific esophagitis (19/69, 27.5%). In contrast to EoE, immunohistochemical assays showed no significant increase in inflammatory infiltration as compared to GERD and controls with the exception of lymphocytic infiltrates in lymphocytic esophagitis. The typical Th2 response in EoE was absent in all EoE variants. The authors
confirmed that all the EoE variants were clinically and histologically active despite the absence of esophageal eosinophilia . EoE variants appear to constitute a spectrum of diseases with classic EoE being the most common and prominent phenotype. In 2023, Salvador Nunes et al. described the different forms of EoE and distinguished between EoE-like esophagitis, lymphocytic esophagitis as a variant of EoE, non-specific esophagitis, and mast cell esophagitis. EoE-like esophagitis was defined as follows: The presence of 0–59 eos/mm2 (<15 eos/hpf) in esophageal biopsies with the presence of typical histopathological features of EoE, particularly dilated intercellular spaces and basal zone hyperplasia. Unlike the remaining two variants, EoE-like esophagitis presents with the highest risk of progressing to EoE over time. For this reason, regular follow-ups with endoscopy and esophageal biopsies are recommended. In addition, EoE-like esophagitis presents with the highest percentage of subepithelial eosinophils. Patients with EoE-like esophagitis also present with a higher incidence of fibrosis as compared to patients with other variants of EoE. The pathogenesis of EoE-like esophagitis remains unclear, and appropriate diagnostic criteria also need to be established. Lymphocytic esophagitis is diagnosed on the basis of the following diagnostic criteria: the presence of inflammation with a predominance of lymphocytes including large counts of intraepithelial lymphocytes (≥30/hpf) accumulated mainly in the peribronchiolar fields, dilatation of intercellular spaces and absence of intraepithelial granulocytes. In contrast to EoE, which is more common in younger males, lymphocytic esophagitis is more common in older women. Clinical presentation is similar to EoE as dysphagia is the most common symptom. Endoscopic changes and treatment methods are also very similar to those observed in EoE. However, it is not yet clear whether lymphocytic esophagitis can be classified as a disease related to (i.e. a variant of) EoE or whether it is a separate and independent disease entity. Further studies are required to unequivocally answer this open question. Nonspecific esophagitis has been defined as histologically confirmed infiltration of lymphocytes or neutrophils that does not meet the diagnostic criteria for lymphocytic esophagitis .
- The findings should be discussed in the context of existing literature, particularly the role of environmental exposures and genetic predisposition.
We have completed this data
- The conclusion should explicitly state the clinical implications of the findings.
Conclusions: All patients with EoE presented with multimorbidities were diagnosed with at least one allergic disease in addition to EoE. Positive staining for eotaxin-1 and negative staining for desmoglein-1, observed in patients with upper gastrointestinal symptoms and allergy who did not meet the diagnostic criteria for EoE, may indicate a subclinical course of the disease. This phenomenon could be attributed to the widespread use of corticosteroids among these patients. However, EoE likely manifests in different variant forms, with the “classic” form representing the most severe phenotype. Awareness of the potential presence of various phenotypes of eosinophilic esophagitis in patients with allergies and gastrointestinal symptoms may aid specialists in the accurate diagnosis and management of these patients.
- The potential for immunohistochemical markers (CCL-11, DSG-1) to serve as early diagnostic indicators should be cautiously interpreter.
Indeed, further studies are undoubtedly required to prove the usefulness of biomarkers such as CCL-11 and DSG-1 in the diagnosis of eosinophilic esophagitis.
Minor comments:
- There are multiple typographical errors throughout the text, e.g., "Symtpoms" should be "Symptoms." A thorough proofread is necessary.
- Abbreviations such as "ICS" and "InCS" should be defined at first mention.
- The reference list is extensive but contains inconsistencies in formatting. Ensure uniform citation style.
- Figures and tables should be labeled clearly and referenced in the text appropriately.
- Some sentences are overly long and difficult to follow. Consider breaking them down for better readability.
We have corrected the errors and implemented the suggestions.

Round 2
Reviewer 1 Report
Comments and Suggestions for Authors
Dear Editor,
I have reviewed the revised manuscript.
The authors have appropriately addressed the previous concerns, and I have no further comments.
Author Response
Dear Sir/Madam,
Thank you very much for your time and the valuable feedback provided in the review.
With kind regards,
The Authors' Team
Reviewer 2 Report
Comments and Suggestions for Authors
The authors have responded fully and comprehensively to all comments, which is why the paper can be considered for publication.
Author Response
Dear Sir/Madam,
We sincerely appreciate your time and the positive evaluation of our work.
With kind regards,
The Authors' Team
Reviewer 3 Report
Comments and Suggestions for Authors
The authors responded "We have not expected obtained results, this is an unexpected find and is merely speculation/suspicion, not a hard conclusion". Their findings are too complex to interpret clearly. so I wonder how much valuable their data are in clinical practice of eosinophilic esophagitis.
Comments on the Quality of English LanguageI wonder whether the authors had received a professional English editing for this revised version.
Author Response
Dear Sir/Madam,
We sincerely appreciate your time and the valuable comments provided in your review.
Eosinophilic esophagitis (EoE) is undoubtedly a disease with a complex and not yet fully understood etiology and pathomechanism. Current diagnostic criteria primarily focus on histopathological examination results. However, there is an urgent need to identify more sensitive and specific biomarkers that could aid in confirming the diagnosis of this condition.
Recent studies suggest that EoE is not a homogeneous disease but rather comprises several distinct phenotypes, with "classical" EoE representing its most extreme form. In our study, we investigated the occurrence of EoE in a group of patients with allergies and upper gastrointestinal symptoms. We approached our research as an exploratory study—an experiment to some extent. The use of additional immunohistochemical staining for eotaxin and desmoglein allowed us to identify a group of patients with a probable EoE phenotype. Although histopathological examination did not confirm EoE in this subgroup, eotaxin staining was positive, whereas desmoglein staining was negative.
We consider our study a contribution to further research, as its findings may be valuable in identifying additional phenotypes of the disease. Further investigations are necessary to define specific biomarkers for different EoE phenotypes. One of our assumptions was the potential influence of glucocorticosteroid therapy in patients with allergic rhinitis (AR) and asthma. A key hypothesis we explored was that the swallowed fraction of these medications might exert a therapeutic effect on EoE. We are aware that the study has limitation and have described them extensively in the discussion section. The corrected version of the manuscript has been reviewed by the professional language editing service.
With kind regards,
The Authors' Team

Reviewer 4 Report
Comments and Suggestions for Authors
The revised manuscript does respond comprehensively to all the major and minor points raised. This revised version is greatly improved. The study offers meaningful insights into the evolving understanding of EoE and its phenotypes, particularly in allergic individuals.
Author Response
Dear Sir/Madam,
Thank you for your time and positive evaluation of our work.
With respect,
The Authors' Team
Round 3
Reviewer 3 Report
Comments and Suggestions for Authors
Although the authors attempted to improve their manuscript, it is still immature to be accepted for publication.
- Although the authors explained IRS scale for the objective assessment of the immunohistochemical reaction intensities, how to determine positive/negative staining for CCL-11 and DSG-1 was not clearly described in Materials and Methods. Without correct and objective cut-off thresholds for positive/negative staining for Eotaxin-1, a chemokine with chemotactic and eosinophil-activating effect and Desmoglein-1, a cadherin involved in the formation of desmosomes, this study cannot be validated appropriately.
- I consider that the group of patients without a confirmed EoE diagnosis but with positive staining for CCL-11 and negative staining for DSG-1 is a key in this study, whereas this group was neither clearly characterized nor investigated as a subclinical course or a subgroup of EoE.
Author Response
Dear Sir/Madame,
We sincerely thank you for your time and the valuable feedback provided in the review.
The method for determining positive/negative staining results for the antibodies has been clearly defined, along with the cutoff thresholds. Each antibody in the methodology has been thoroughly and precisely described (manufacturer, clone, catalog number). For each antibody, control testing was performed according to the methodology provided by the manufacturer – for example, for Desmoglein-1, the control was skin, where the manufacturer provides detailed instructions for the reaction procedure and its expected result.
The indicated result serves as a reference point for our staining procedures—any preparations with lower intensity were assigned fewer points, which is why this is a semiquantitative scale. Similarly, the localization of the reaction (cytoplasmic, nuclear, membranous) was assessed in relation to the specific antibody. Therefore, we used antibodies from different manufacturers as we aimed to obtain comparative results on human tissues. In the methods and materials section, we do not provide detailed staining protocols, only the antibody information, which allows for easy and precise identification of the relevant protocol.
We fully agree with your opinion that the group of patients with allergy and symptoms related to the upper gastrointestinal tract, as well as positive immunohistochemical results for eotaxin and negative results for desmoglein, is the most interesting. Although histopathological examination did not confirm EoE in patients from this group, we believe that the positive staining for eotaxin may indicate eosinophilic inflammation, while the negative staining for desmoglein could suggest damage to the integrity of the epithelial barrier. Therefore, we likely are dealing with one of the phenotypes of EoE in this case. This group consisted of 47 patients. In this subgroup, 14 patients were on inhaled corticosteroids (ICSs), 44 patients were taking intranasal corticosteroids (InCSs), and 8 patients were on both ICSs and InCSs. One patient was receiving systemic corticosteroids. Among these patients, 85% were allergic to grasses, 55% to house dust mites, 66% to birch, 68% to alder, and 64% to hazel. 10% were allergic to wheat and milk, and 9% to hen's egg. Additionally, all patients in this group were diagnosed with allergic rhinitis, 10 had asthma, 4 had skin allergies, and 30 had food allergies. Endoscopic changes in this group were not severe, as the esophagus appeared normal in most cases, with only 4 patients showing erosions and 2 having rings. Of course, to confirm that we are dealing with an EoE phenotype in this case, further hard evidence is needed, and additional studies are required, especially given that there are currently no reliable guidelines necessary for the diagnosis of these disease variants.
With kind regards,
The Authors' Team